# SteerVLA: Steering Vision-Language-Action Models Toward Effective Long-Tail Driving

## Abstract

A fundamental challenge in autonomous driving is the integration of high-level, semantic reasoning for long-tail events with low-level, reactive control for robust driving. While large vision-language models (VLMs) trained on web-scale data offer powerful common-sense reasoning, they lack the grounded, embodied experience necessary for safe vehicle control. Conversely, policies trained on driving data exhibit strong reactive skills, but often fail in novel scenarios that require abstract understanding. We posit that an effective autonomous agent must leverage the world knowledge of VLMs to steer a grounded driving policy, rather than attempting to embed all knowledge into a single monolithic model. To this end, we propose "SteerVLA", a hierarchical driving policy composed of a high-level VLM planner and a low-level vision–language–action (VLA) policy. The planner produces fine-grained language commands, which steer a flexible, low-level policy for control. To train these policies, we leverage VLMs to augment existing real-world and simulation data with dense annotations in hindsight, which we find is essential for strong reasoning and steerability. We evaluate SteerVLA in challenging closed-loop long-tail scenarios, where it outperforms state-of-the-art methods.

## 1 Introduction

Despite rapid progress in autonomous driving systems, long-tail scenarios remain particularly challenging due to their inherent scarcity in driving data and the complex reasoning they require. A truly autonomous vehicle must handle ambiguous traffic flow in construction zones, unpredictable pedestrian behavior, and blocked lanes due to accidents, as well as compositions of these scenarios that may occur sequentially or in combination. Figure 1, for example, depicts a car suddenly merging from being parked into the driving lane in front of the ego vehicle. Meanwhile, a traffic cone blocks the road, reducing a two-lane road to one shared lane. Handling these long-tail scenarios effectively is fundamentally important to build safe and robust driving systems (Tian et al., 2024).

Vision-Language-Action (VLA) models, derived from Vision-Language Models (VLMs) fine-tuned via imitation learning, leverage strong semantic priors to generate embodied actions (Brohan et al., 2023a; Kim et al., 2024; Zhou et al., 2025a). However, directly fine-tuning a VLM or VLA on driving data does not guarantee that it will retain the general knowledge necessary to generalize to long-tail scenarios, as the model may fail to retrieve information stored in its weights (Miller & Matzel, 2006) or even unlearn knowledge acquired during pretraining (Yao et al., 2024; Driess et al., 2025).

To address these limitations, we present SteerVLA, a novel framework to obtain VLA-based driving policies that remain effective both under normal conditions and in long-tail driving scenarios. Our key insight (Figure 1) is to decompose the driving problem into two stages, which retain the powerful reasoning capabilities in pretrained VLMs and the fine-grained control actions from specialized driving models.

Specifically, we construct two components: 1) a high-level planner that performs semantic and common sense reasoning to analyze driving scenarios, based on camera images, routing commands (e.g., "Turn left") from navigation APIs and historical vehicle states. This model then outputs reasoning traces and meta-actions. 2) a low-level policy that generates precise control actions based on the meta-actions. Here, the key technical challenge we solve is enabling the high-level planner

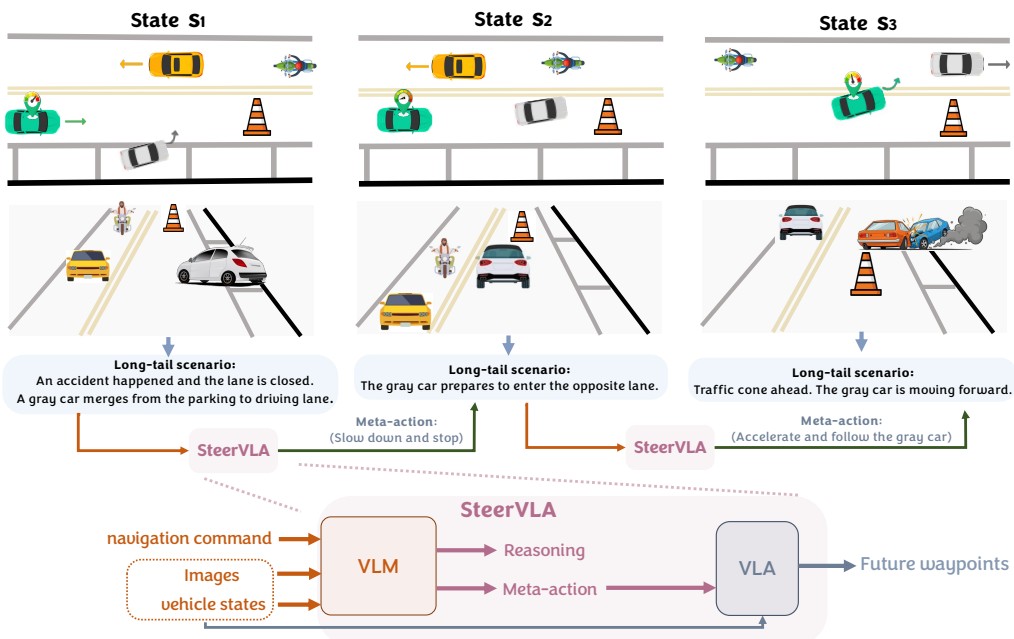

Figure 1: **Overview**. SteerVLA enables effective reasoning in long-tail driving scenarios. For example, in $S_1$, a car suddenly merges into the driving lane, requiring a preemptive slowdown and stop. In $S_2$, some vehicles remain in the opposite lane, and the car should hold its position and wait for the leading vehicle's action. In such long-tail scenarios, SteerVLA performs intensive reasoning over the driving context to generate correct control actioncontrol actions.

to reason over the driving scenarios correctly. For example, when an accident blocks the road, and the front car enters the opposite driving lane, the high-level planner should reason that "*It appears that the lane is now shared in both directions since the front car is moving. The vehicle should proceed while observing cautiously*" and output a conservative acceleration action from a stopped state. Moreover, given camera images and vehicle states, the low-level policy should map the general meta-actions to precise control actions such as speed control and steering angle.

Difficulty arises from the lack of supervision on the high-level planner's meta-action and the lack of grounded datasets that contain natural-language meta-actions and corresponding precise control actions. Therefore, to provide high-quality supervision for both stages, we additionally design an automatic data generation pipeline which, given realistic driving scenes, guides a VLM to simulate long-tail driving scenarios and generate grounded meta-actions and control actions. Furthermore, these meta-actions are refined from generic commands, such as "turn left" and "go straight" with additional information from the trajectory to better steer the low-level policy, resulting in commands like "turn left aggressively, accelerating slightly" and "go straight, nudging right slightly, driving normally". We finetune the high-level planner on these high-quality data to correctly reason over complex scenes, ensuring well-grounded meta-actions. For the low-level policy, we adopt a classifier-free guidance strategy and train the model to better follow the meta-actions, obtaining precise control actions.

We evaluate SteerVLA on the Bench2Drive (Jia et al., 2024) benchmark in the CARLA (Dosovitskiy et al., 2017) simulator. We demonstrate that SteerVLA outperforms a state-of-the-art baseline on this benchmark, and achieves notably strong results on long-tail scenario categories in the multi-ability assessment. To summarize, our contributions are:

- SteerVLA, a hierarchical training framework for driving policies that exhibit strong reasoning capabilities and driving performance, especially in long-tail scenarios.

- A data generation pipeline that can generate refined language labels to improve instruction following.

## 2 RELATED WORK

**End-to-end driving policies.** While autonomous driving has traditionally consisted of methods that use a stack of perception, prediction, and planning modules (Hu et al., 2023; Huang et al., 2021; Sun et al., 2021), massive progress has been made with end-to-end imitation learning methods that directly map multi-modal inputs to driving commands (Feng & Alahi, 2025; Nguyen et al., 2025; Zheng et al., 2025; Hegde et al., 2025). These methods generally excel in generic driving scenarios, but generalizing to long-tail scenarios is challenging without any inherent semantic knowledge about driving. As a result, these scenarios must be well covered in the training data, but as they are rare, this is not the case. Some methods use world-modeling (Bartoccioni et al., 2025; Gao et al., 2024; Russell et al., 2025) to understand the consequences of different actions without simulation, leveraging this information to learn safe, controllable driving. While these methods are aligned with tackling long-tail scenarios, they require an immense amount of data to train, and it is unclear how well they can model rare scenarios. SteerVLA leverages a VLM backbone for the high- and low-level components, which allows it to inherit the semantic reasoning capabilities and general vision-language priors from VLM pre-training.

**LLM- and VLM-based driving policies** Several works have gone beyond training end-to-end policies from scratch and leverage large language and vision-language models to adopt their pre-trained capabilities. Various works leverage pre-trained large language models, fine-tuning them on driving data (Jia et al., 2023; Yuan et al., 2024; Hwang et al., 2024b; Arai et al., 2025; Zhou et al., 2025a; Fu et al., 2025; Gao et al., 2025; Zhou et al., 2025c). LLM-Driver (Chen et al., 2024) and DriveGPT4 (Xu et al., 2024b) integrate multimodal inputs with foundation models to enhance driving performance. GPT-Driver (Mao et al., 2023a), Agent-Driver (Mao et al., 2023b), and AgentThink (Qian et al., 2025) adapt ChatGPT as a motion planner through text-based fine-tuning. DriveMLM (Wang et al., 2023) and LMDrive (Shao et al., 2024) propose end-to-end closed-loop driving models with LLM backbones, though their ability to follow human instructions remains limited. Inspired by the success of pretrained vision-language models (VLMs), several works have introduced *vision-language-action* (VLA) models (Brohan et al., 2023a), which typically consist of a VLM backbone fine-tuned to produce robot actions conditioned on visual inputs and language instructions (Kim et al., 2024). These models benefit from excellent cross-modal grounding between language and vision, enabling the transfer of internet-scale semantic knowledge from the pretraining data. However, a key challenge for these methods is retaining the strong capabilities learned during pre-training that can be destroyed when transferring to the domain of action prediction, a task very different from those found in VLM pre-training. As a result, we use a hierarchical model that allows the high-level policy training to stay closer to the VLM pre-training tasks, resulting in better reasoning and less overfitting to the training data. The low-level policy is trained with detailed language instructions, which helps induce better language following and allows training to target action prediction.

**Reasoning in autonomous driving.** Recent works have sought to imbue VLAs with reasoning capabilities (Zawalski et al., 2024; Zhao et al., 2025; Mu et al., 2023; Shi et al., 2024; Belkhale et al., 2024; Chen et al., 2025) to improve generalization and compositional task-following, and have introduced hierarchical structures to enable robust long-horizon behavior (Black et al., 2024; Intelligence et al.). Despite these advances, many VLAs tend to overfit to the training distribution, eroding the original capabilities of the underlying VLM — a phenomenon we term *knowledge collapse*. To mitigate this, we employ distinct models connected via a carefully designed interface, preserving pre-trained knowledge while ensuring effective knowledge insulation. Some efforts leverage pretrained VLMs to provide driving systems with broad world knowledge and reasoning capabilities (Sima et al., 2024; Xu et al., 2024a), while others develop VLA policies by fine-tuning VLMs with an action head (Hwang et al., 2024a; Zhou et al., 2025b; Tian et al., 2024; Renz et al., 2025). These works typically focus on low-level driving, with language used primarily as an auxiliary signal or structured instruction. Related research has explored different ways to generate language supervision, including chain-of-thought (CoT) reasoning to detect agents and describe the scene, casting driving as VLM tasks such as QA or captioning, leveraging counterfactual data, and providing explainability or justification signals. While these methods improve reasoning or generalization, they remain largely descriptive. In contrast, we augment data with dense, prescriptive *meta-actions* that specify the vehicle's next behavior based on the scenario, enabling SteerVLA to follow open-ended user instructions in a steerable and interpretable manner. Most similar to our work is Simlingo, a unified model trained with CoT rather then a hierarchical architecture. This

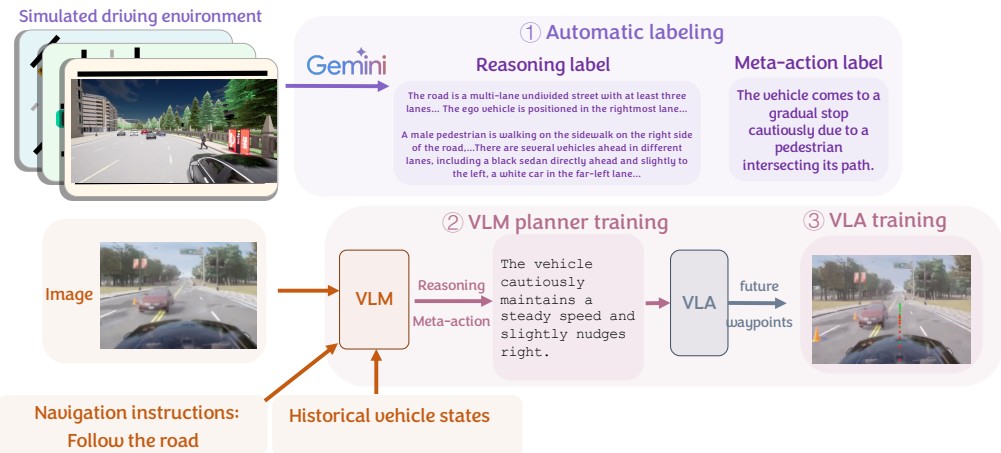

Figure 2: **Framework overview**. Our model is a hierarchical driving policy where a high-level planner generates reasoning traces and detailed meta-actions, which guide a low-level VLA specialized in fast and accurate waypoint prediction. This design improves generalization to long-tail scenarios by offloading complex reasoning to the planner while keeping the control loop efficient and robust. To generate meta-action labels, we propose an auto-labeling pipeline that produces fine-grained action descriptions and reasoning traces to enhance dataset quality.

method also focuses on long-tail driving scenario capabilities and achieves state-of-the-art performance on the Bench2Drive benchmark. However, SimLingo relies on access to "action dreaming" data, generated in the CARLA simulator with access to privileged information to improve reasoning and steering capabilities. SteerVLA is trained without data that requires privileged simulation information and uses labeling methods that are transferable from simulation to real-world data, making it easier to scale and extend to real driving scenarios. As well, we do not rely on additional data to improve reasoning and steerability, but achieve this through our hierarchical architecture and detailed meta-action labels to steer the low-level policy.

## 3 STEERVLA

We formulate autonomous driving as a sequential decision-making problem. At each timestep $t$, the agent receives an observation $o_t = \{I_t, q_{t-k:t}, \ell_t\}$, where $I_t$ is the current front-view camera image, $q_{t-k:t}$ denotes the recent history of ego-vehicle proprioceptive states (e.g., past speeds and headings over the last $k$ steps), and $\ell_t$ is a routing command provided by a navigation system (e.g., turn-by-turn guidance such as "turn left in 50 m"). The objective is to predict a chunk of future actions $A_t = [a_t, a_{t+1}, \ldots, a_{t+H-1}]$ that specify low-level control signals (e.g., future waypoints) over a horizon $H$ (Zhao et al., 2023). A driving policy is therefore a conditional distribution $\pi(A_t \mid o_t)$ that maps the current observation and routing command to a distribution over action chunks. Training proceeds by maximizing the likelihood of expert demonstrations: $\max_\theta \mathbb{E}_{(A_t, o_t) \sim D} [\log \pi_\theta(A_t \mid o_t)]$, where $D$ is a dataset of expert driving trajectories paired with synchronized routing commands.

We focus on the challenge of long-tail driving scenarios, where rare and unanticipated events require strong generalization and common-sense reasoning from the policy. VLAs are a strong backbone for driving because they combine semantic grounding from vision–language pretraining with imitation learning from driving domain data. To alleviate the challenge for a single model to handle both complex reasoning and low-level control reliably, we introduce a hierarchical decomposition. A high-level planner first reasons about the scene and routing context to produce a *meta-action*—a detailed driving instruction for the ego vehicle movement (e.g., "yield, then turn left into the near lane; keep speed $\leq 25$ km/h"), accompanied by a short reasoning trace $\tau_t$ that records its situational justification. Formally, given observation $o_t = \{I_t, q_{t-k:t}, \ell_t\}$ with image $I_t$, proprioceptive history $q_{t-k:t}$, and routing command $\ell_t$, the high-level planner outputs $l_{ma} = (\tau_t, m_t) \sim \pi_{hl}(l_{ma} \mid o_t)$. The low-level VLA then predicts future waypoints $A_t = [a_t, a_{t+1}, \ldots, a_{t+H-1}]$ conditioned on both the observation and the meta-action, i.e., $A_t \sim \pi_{ll}(A_t \mid o_t, m_t)$. This design improves generalization by offloading high-level reasoning to the high-level planner, while allowing the VLA to specialize in

fast and accurate waypoint prediction. It also supports flexible composition across vehicle platforms and efficient deployment, since the larger high-level planner runs infrequently while the lightweight VLA operates continuously at control.

To train such a system, we use ego-centric, detailed meta-actions as the interface between the high- and low-level policies. These meta-actions are generated by leveraging additional information from the vehicle's trajectory (e.g., speed, compass heading) in hindsight, either creating them from scratch or refining existing labels. This approach enables the planner to retain strong reasoning skills while grounding its decisions in a robust low-level policy through an information-rich language interface.

### 3.1 STEERVLA'S HIERARCHICAL ARCHITECTURE

A hierarchical architecture offers three fundamental advantages that allow us to train a capable driving policy for complex driving scenarios.

First, each policy can specialize in a part of the driving task. The high-level planner is trained to perform a semantic reasoning task, that is, to use the image observation, history, and language instruction to generate a reasoning trace, which describes the scene, other agents, and various other factors that might impact driving behavior and the meta-action that describes the action the vehicle should take. This task is much closer to the VLM pre-training tasks, such as question-answering and generic reasoning tasks, easing the distribution shift during finetuning and preventing loss of the original VLM capabilities (Driess et al., 2025). We can then train the low-level policy to be steerable and focus on fine-grained driving skills rather than reasoning. An overview of the architecture is provided in Figure 2.

Second, this architecture gives us the flexibility to train on different data sources. The high-level planner can be trained more easily on cross-domain data, including real-world and simulated data, as it interacts with concepts that are transferable across these domains (e.g., vehicles, roads, and pedestrians in the real world and simulation can be reasoned about in the same way, even though they visually and dynamically may differ). The low-level policy can be treated as a domain specialist and can be trained only on simulated or real-world data.

Last, while a hierarchical model consists of two models, making it more expensive to run at every step, we can query the high-level planner and low-level policy at different frequencies, allowing us to scale up the size of the policy overall (about 7 billion parameters) while not requiring us to sacrifice inference time at every step.

**High-level planner.** We finetune the high-level planner with Gemma3-4B (Team et al., 2025) as the base model, a small but powerful VLM, leveraging its strong semantic priors to generate a suitable meta-action that captures both the nuances of the suitable driving behavior for the scenario. We structure the query to the VLM as a visual question-answering problem by providing the current visual observation together with a visual frame from 0.5 seconds prior, a six second long history of ego states (speed and heading) sampled at 2 Hz, and a routing command. We train the model via a next-token prediction objective to generate a chain-of-thought reasoning trace describing the positions and movements of critical objects and agents in the scene, followed by an appropriate meta-action.

**Low-level VLA policy.** Once a meta-action has been generated, the steerable low-level policy predicts actions that align with the desired behavior. To this end, we train a meta-action-conditioned VLA policy (see Section 3.2 for details on generating meta-action labels) using PaliGemma (Beyer et al., 2024) as the pre-trained backbone for the VLA. We follow the recipe from (Kim et al., 2024), and overwrite rarely used or outlier tokens to represent each bin in a discretized action space where each dimension contains 512 uniform bins. We train on actions normalized to the range -1 to 1 based on the dataset statistics (Octo Model Team et al., 2024; Brohan et al., 2023b). We train our policy to predict Cartesian coordinate waypoints in the xy-plane. The policy takes as input the current front camera image observation of the vehicle and the current speed, but we do not include history to avoid causal confusion at the action generation level (Torne et al., 2025). Unlike OpenVLA, our model generates an open-loop *action chunk* (Chi et al., 2023; Fu et al., 2024) which enables smooth temporally correlated actions and decreases compute requirements. We auto-regressively generate this action chunk with a horizon of 10 time steps with a fixed frequency of 4 Hz, resulting in a total of 20 tokens predicted.

## 3.2 GENERATING LANGUAGE LABELS FOR STEERVLA

Many driving datasets lack fine-grained human-annotated labels. In order to produce detailed and accurate natural language labels of vehicle actions, we leverage trajectory and course information, as well as the vehicle's observations. We begin by splitting our trajectories into short 2-5 second chunks, creating splits based on the angular velocity and acceleration trends of the vehicle. Using the camera intrinsic and extrinsic properties, we project the future trajectory taken by the vehicle onto a first-person front-facing camera view. We then perform a two-stage query to a VLM: we begin by providing the aforementioned projection, lane identification information, and the vehicle's velocity and heading over the duration of the chunk. We query the VLM to categorize the baseline action taken by the vehicle over the duration of this chunk (e.g., changing lanes, turning, or continuing forward). We then perform a refinement step to determine the style and motion extent of the driving behavior by providing the VLM with the vehicle's ego states, which include its speed and course, over time, as well as the previously produced baseline label, to produce a nuanced description of the vehicle's action over the duration of the chunk. For example, we transform the original label "the car rolls through the stop sign" into the more fine-grained "the car rolls through the stop sign with a slight right turn, accelerating gradually, driving normally". This refinement step is crucial for passing as much information as possible to the low-level policy and can be applied to any existing language-labeled driving dataset, allowing us to augment this data with additional information that can improve steerability and performance. We additionally generate reasoning traces for each of the trajectories in the training data that describe the scene and analyze the motion of other agents. We train the high-level to generate these reasonings as an auxiliary task, and only provide the meta-action as input to the low-level policy. More details on the prompts used for our autolabeling pipeline are provided in Section A3 of the appendix.

## 4 EXPERIMENTS

We focus our experiments on evaluating the long-tail scenario reasoning capabilities of SteerVLA. To this end, we evaluate SteerVLA in closed-loop settings to measure its raw driving performance and its ability to tackle unusual cases, such as navigating a single lane when the rest of the road is closed for construction, control loss, and varying weather conditions. We also evaluate SteerVLA open-loop on real-world datasets, including BDD-X (Kim et al., 2018) and NuScenes (Caesar et al., 2020). While SteerVLA is competitive with state-of-the-art methods, these results are less informative than closed-loop performance, as they can only give a local estimate of the performance of a policy and often use metrics like L2 error, which cannot accurately capture the nuances of the policy's behavior. These results are provided in Section A1 of the appendix.

**RQ1:** How effective are SteerVLA's reasoning capabilities in closed-loop driving scenarios?

**RQ2:** How does SteerVLA compare to prior state-of-the-art methods in long-tail scenarios?

**RQ3:** How do design decisions contribute to a capable high-level planner for SteerVLA?

**RQ4:** Does SteerVLA's detailed language augmentation pipeline improve language following and steerability?

### 4.1 EXPERIMENTAL SETUP

The majority of our experiments use the CARLA simulator to perform closed-loop evaluation of SteerVLA. We use the commentary labels from the Simlingo (Renz et al., 2025) driving dataset, and once again apply our data augmentation pipeline to generate reasoning traces. We train the high-level planner on this dataset and additionally train the low-level policy on a mixture of the SimLingo data and a dataset we collected in CARLA, which includes diverse driving scenarios, mostly focusing on weather and lighting variation, generated using the built-in traffic agent autopilot. We additionally modify our action space such that we not only predict actions with a fixed time interval (every 0.25 seconds) but also generate xy-waypoints at a fixed distance, such that we can use the same waypoint controller as SimLingo. We run the high-level planner at 5 Hz and the low-level policy at 20Hz. The high-level planner has an inference latency of 0.55s, and the low-level policy has a latency of 0.69s. While we did not optimize inference for these models, as we evaluate them in simulation, we plan to reduce inference time with techniques such as KV caching for real-world deployment. We

evaluate SteerVLA on the Bench2Drive (Jia et al., 2024) benchmark, which contains 220 driving scenarios in 12 towns, including adverse weather and lighting conditions, such as fog, nighttime driving, and various long-tail driving scenarios, such as a road being reduced to a single two-way lane, construction sites, emergency vehicles, and other multi-agent reasoning problems.

**Baselines.** We evaluate several recent vision-language-action (VLA) baselines. 1) SimLingo (Renz et al., 2025), a vision-only VLM framework that addresses closed-loop driving, vision-language understanding, and language-action alignment, relying solely on cameras and avoiding costly sensors such as LiDAR. SimLingo additionally leverages "action-dreaming" data, which is counterfactual data used to improve its language following capabilities. SimLingo is currently the top method on the CARLA 2.0 leaderboard. 2) DriveMoE (Yang et al., 2025), built upon the $\pi_0$ foundation model (Black et al., 2024), employs a mixture-of-experts architecture with a scene-specialized vision MoE and a skill-specialized action MoE to achieve adaptive decision making for autonomous driving. 3) ORION (Fu et al., 2025), a holistic E2E framework that integrates a QT-Former for long-term history aggregation, an LLM for driving scenario reasoning, and a generative planner for precise trajectory prediction. ORION further aligns reasoning and action spaces, enabling unified optimization across both planning and visual question answering, though at the cost of greater complexity and computational demand. 4) AutoVLA (Zhou et al., 2025c), which enhances a pretrained VLM with a physical action codebook for vehicle motion, effectively bridging semantic reasoning and low-level control.

## 4.2 EVALUATING STEERVLA ON DRIVING PERFORMANCE

Towards answering **Q1** and **Q2**, we evaluated SteerVLA on the Bench2Drive (Jia et al., 2024) benchmark, which contains 220 different routes with scenarios ranging from merging and overtaking to navigating a single two-way lane in heavy traffic.

| Method | Sensors | DS ↑ | SR(%) ↑ | Ability (%)↑ | | | | | |
|---|---|---|---|---|---|---|---|---|---|
| | | | | Merging | Over-taking | Emergency Brake | Give Way | Traffic Sign | Mean |
| DriveMoE | M | 74.22 | 48.64 | 34.67 | 40.00 | 65.45 | 40.00 | 59.44 | 47.91 |
| ORION | M | 77.74 | 54.62 | 25.00 | **71.11** | **78.33** | 30.00 | 69.15 | 54.72 |
| AutoVLA | M | 78.84 | 57.73 | - | - | - | - | - | - |
| SimLingo | S | 85.94 | 66.82 | 57.50 | 60.00 | 76.67 | 50.00 | 73.16 | 63.46 |
| SteerVLA (Ours) | S | **86.81** | **69.55** | **66.25** | 61.36 | 76.67 | **80.00** | **76.32** | **72.12** |

Table 1: **Evaluation of SteerVLA on Bench2Drive.** Metrics include Driving Score (DS), Success Rate (SR%), and specialized abilities (Merging, Overtaking, Emergency Brake, Give Way, Traffic Sign) with overall Mean performance. Compared to the state-of-the-art, SteerVLA outperforms the best performing baseline (Simlingo) and specifically excels in merging maneuvers, overtaking, and traffic sign recognition. M/S refers to Multi-camera/Single camera.

## 5 RESULTS

We find that SteerVLA has strong performance on the Bench2Drive benchmark, achieving better performance than SimLingo in success rate and driving score. We observe that SteerVLA tends to outperform SimLingo in highly dynamic scenarios (i.e., a vehicle turning into the same lane as the ego vehicle, or a vehicle door suddenly opening in the path of the ego vehicle). SteerVLA's reasoning trace structure guides the policy to make conjectures about the movement intent of the other agents within the scene, enabling SteerVLA to prepare and preemptively react to adversarial behavior. The multi-ability scores further elucidate the advantages of SteerVLA, as SteerVLA outperforms all baselines on driving score and success rate, and achieves a high ability score in 3 of the 5 categories.

However, we still observe failure cases for SteerVLA, which mainly arise from hallucination of the high-level planner. Two such cases, which are especially impactful, are the incorrect detection of whether a vehicle is stationary and the state of critical scene elements, such as traffic lights,

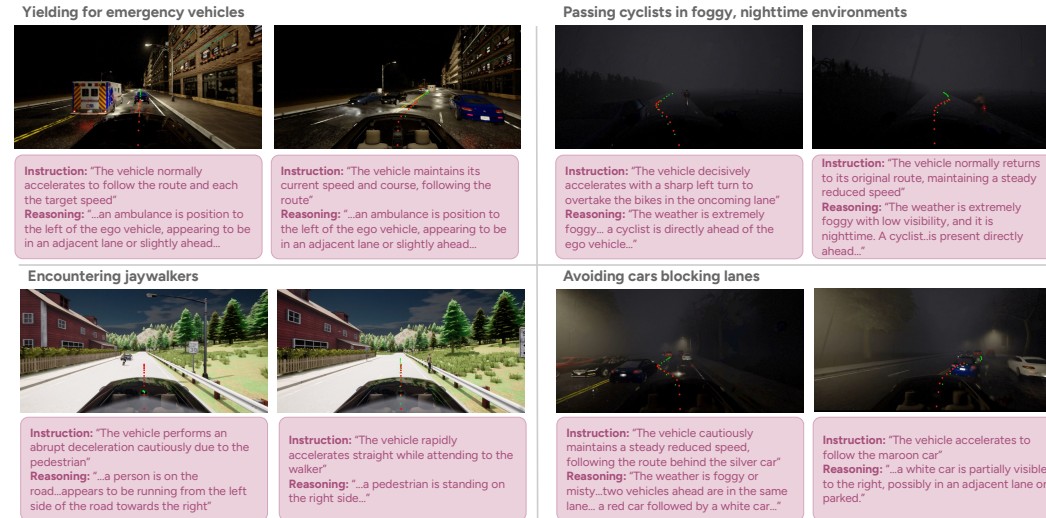

Figure 3: **Example rollouts of SteerVLA in long-tail scenarios.** SteerVLA can handle a broad set of scenarios, including complex interactions with multiple agents, such as cyclists and pedestrians, and dealing with non-typical behavior.

in low-visibility weather. These cases could lead to unsafe behavior; however, as our low-level policy is trained on safe data, it will usually reject unsafe driving meta-actions and still perform safe behaviors.

## 5.1 ABLATIONS OF STEERVLA'S HIGH-LEVEL POLICY

| Method | Base Model (# params) | Architecture | DS ↑ | SR(%) ↑ |
|---|---|---|---|---|
| SimLingo (Renz et al., 2025) | InternVL-2 (1B) | unified (CoT) | 85.94 | 66.82 |
| SteerVLA | PaliGemma3 (3B) | unified | 67.66 | 43.64 |
| SteerVLA | PaliGemma3 (3B) | unified (CoT) | 55.48 | 34.65 |
| SteerVLA | InternVL-2 (1B) | hierarchical | 70.60 | 46.05 |
| SteerVLA | Gemma3 (4B) | hierarchical | **86.81** | **69.55** |

Table 2: **Ablations of the model architecture of SteerVLA.** The results verify the effectiveness of our hierarchical architecture and performance with different model capacities.

The high-level policy does the bulk of the reasoning in the driving scenarios and, therefore, is most important for strong long-tail scenario performance. Two factors that influence the capability of this model are (i) the base VLM and (ii) the hierarchical structure of our policy.

We compare SteerVLA trained as a unified model with PaliGemma 3B (Beyer et al., 2024) as the base VLM to determine the importance of using the larger Gemma3 backbone, as well as the importance of our hierarchical model. We train two versions of this unified model: 1) a model which directly maps prompts to control actions and does not include intermediate representations, and 2) a CoT version of SteerVLA, which predicts the meta-actions as an auxiliary task. We additionally train a version of the high-level planner using the same base model as SimLingo, InternVL-2, and present the results in Table 2.

We observe that the unified models suffer a dramatic drop in performance compared to their hierarchical counterparts, exhibit lower quality reasoning abilities, and this makes it difficult to properly handle dynamic scenarios. Specifically, we observe that it tends to lose control more easily, which may be a result of having to learn to both reason and perform the control task, two vastly different capabilities.

| Category | Accuracy (%) |
|---|---|
| Accelerating/Decelerating | 96 |
| Turning | 84 |
| Lane changing | 68 |

Table 3: **Accuracy of meta-action labels.** We manually evaluate the correctness of the meta-action labels generated with our auto-labeling pipeline on the NuScenes dataset.

| Label type | DS | SR(%) |
|---|---|---|
| Original | 77.65 | 57.73 |
| Refined | 86.81 | 69.55 |

Table 4: **Ablation of label refinement.** Comparison of SteerVLA's performance with and without refined labels.

We also observe that using the InternVL-2 as the base model for the high-level planner hurts performance, while outperforming the unified models. InternVL-2 is trained a multi-modal reasoning model, focusing primarily on vision and image understanding, while Gemma3 is a powerful, generalist reasoning model. It is possible that InternVL-2 cannot handle the more complex language of the refined labels we train on, meaning that inherent strong reasoning capabilities in the base model is necessary for SteerVLA. The hierarchical architecture of SteerVLA allows it to be more robust, as the high-level planner can focus on the reasoning task and provide the relevant information to the low-level policy to perform the control task. Ultimately, we find that the hierarchical structure of SteerVLA scaffolds the reasoning power needed to determine the appropriate moments to take certain actions and react to other agents within the scene.

### 5.2 ABLATION OF THE META-ACTION LABELS

The high-level planner communicates with the low-level policy through the meta-actions, and therefore, how these meta-actions are designed is essential to the overall performance of SteerVLA.

We study the labels themselves and their accuracy relative to prior methods with human annotators in the loop. We also study the performance of SteerVLA when trained on the original, unaugmented labels from the SimLingo dataset, and refined labels produced with the pipeline described in Section 3.2, shown in Table 4.

We first evaluate a sample of meta-action labels generated for the NuScenes dataset, which does not include pre-existing language labels. Therefore, we apply the entire pipeline described in Section 3.2 to generate these labels. We specifically evaluate the accuracy of the base meta-actions, rather than the fine-grained details added during refinement, as it is easier to evaluate correctness definitively at the base meta-action level. We randomly sample 20 labels from accelerating/decelerating, turning, and lane changing scenarios, and manually evaluate their correctness. These results are provided in Table 3.

We observe that our pipeline can achieve reliable accuracy on classifying turns and acceleration or deceleration. However, our method struggles more on classifying lane changes, which requires the detection of fine spatial and temporal cues, especially when relying on information from a single front-facing camera. Open-loop results on the NuScenes planning benchmark that leverage these labels are included in Section A1 of the appendix, demonstrating that these labels are sufficient for achieving strong performance, competitive with state-of-the-art methods.

We find that our hierarchical architecture and refined labels are highly important to ensure that dense information about the actions is passed to the low-level policy.

## 6 DISCUSSION

We presented SteerVLA, a hierarchical vision-language-action (VLA) model for autonomous driving achieves strong performance in long-tail scenarios. Our approach decomposes the problem into a high-level language-based reasoning step and a low-level action generation step, using structured meta-actions as the interface between them. By doing so, SteerVLA leverages vision-language model (VLM) priors to interpret behavioral instructions in language space before producing raw control actions.

To train this hierarchical policy, we introduce a novel autolabeling pipeline that generates plausible high-level behavior specifications and meta-action annotations from unlabeled self-driving datasets. This enables SteerVLA to respond effectively to complex, unstructured language prompts, *including those unseen during training*.

**Limitations and Future Work.** While our results show improved reasoning and steerability, SteerVLA has several limitations. First, the quality of autolabeling is constrained by the capabilities of the underlying VLM. Although labeling based on video snippets would be ideal, current VLMs still struggle with dynamic, temporally grounded reasoning compared to static scene understanding. In future work, we aim to bootstrap driving-specific video reasoning capabilities into the labeling pipeline.

Besides, we see an opportunity to incorporate techniques such as reinforcement learning from human feedback (RLHF) to improve the alignment of the high-level planner with user preferences and downstream driving behavior. We hope that future extensions of SteerVLA will build upon these directions to enhance its adaptability and human-aligned decision-making.

Additionally, we acknowledge that although we see promise of transfer of SteerVLA to real-world driving, there are still factors to consider for real-world deployment. Real-world drivers will have much more diverse behavior than the rule-based driving agents used in the CARLA simulator. This means that more data will be required to better reason about a driver's behavior. However, we believe that our work provides a good first step into focusing on retaining and refining reasoning capabilities for autonomous driving agents.

## 7 ETHICS STATEMENT

This research focuses on the architectural and algorithmic development of hierarchical autonomous driving policies. Our main contribution is SteerVLA, a framework that integrates vision-language models with driving data to improve reasoning, steerability, and performance in long-tail driving scenarios. This work is methodological in nature and does not introduce new domain-specific ethical claims or societal impacts beyond the general considerations associated with large-scale vision-language models and autonomous driving research. All experiments are conducted in simulation, with no human subjects involved.

## 8 REPRODUCIBILITY STATEMENT

We provide a comprehensive description of the proposed methodology in Section 3, including the task definition and algorithmic structure of SteerVLA. To facilitate reproducibility, we will make the complete code implementation publicly available at `https://anonymous.4open.science/r/steervla-B26D`. Additional details on experimental settings, and data processing are provided in the Appendix. Together, these resources enable independent researchers to reproduce the experiments and analyses presented in this work.

## 9 THE USE OF LLMS

LLMs were used solely as writing assistants to improve clarity, style, vocabulary, and conciseness. All research ideas, experimental design, data analysis, and scientific content remain the authors' original work. All LLM-assisted edits were carefully reviewed to ensure accuracy and fidelity to the intended meaning.

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

## A APPENDIX

### A1 BDD-X OPEN-LOOP RESULTS

We train SteerVLA with the original BDD-X language labels and those that result from the refinement step of our auto-labeling pipeline (see Table 5). We observe that by simply using our architecture, we achieve a massive improvement in performance, dramatically decreasing the RMSE by at least a factor of 2. With the refined labels, we gain an additional decrease in RMSE, demonstrating that they help induce more fine-grained language following, such as the aspects of "style" and "motion extent" described in Section 3.2.

| Method | Speed (m/s) RMSE↓ | Angle (°) RMSE↓ |
|---|---|---|
| DriveGPT4 (Xu et al., 2024b) | 1.30 | 8.98 |
| SteerVLA w/ orig. labels | 0.56 | 2.32 |
| SteerVLA w/ refined labels | **0.53** | **2.16** |

Table 5: **Open-loop comparison of SteerVLA (with original and refined language) to DriveGPT4 on BDD-X.** SteerVLA outperforms DriveGPT4 in speed and turning angle prediction.

| Method | Traj L2 (m) ↓ | | | |
|---|---|---|---|---|
| | 1s | 2s | 3s | Avg. |
| TOKEN (Tian et al., 2024) | 0.36 | 0.70 | 1.46 | 0.81 |
| PARA-Drive (Weng et al., 2024) | 0.26 | 0.59 | 1.12 | 0.66 |
| DiMA+(VAD-Base) (Hegde et al., 2025) | 0.18 | 0.48 | 1.01 | 0.56 |
| Agent-Driver (Mao et al., 2023b) | 0.16 | 0.34 | 0.61 | **0.37** |
| SteerVLA (Ours) | 0.18 | 0.39 | 0.63 | 0.40 |

Table 6: **Open-loop comparison of SteerVLA on the NuScenes planning benchmark.** SteerVLA achieves the lowest overall L2 error compared to state-of-the-art methods.

## A2 NUSCENES PLANNING BENCHMARK OPEN-LOOP RESULTS

We also perform the full auto-labeling pipeline on the NuScenes dataset and evaluate SteerVLA against a set of baselines on the NuScenes planning benchmark (see Table 6). We run the policy at 2 Hz, getting the L2 error over horizons of 1, 2 and 3 seconds. SteerVLA outperforms significantly outperforms the majority of the baselines. Specifically, SteerVLA still achieves a low L2 error, even with a horizon of 3 seconds, whereas TOKEN, PARA-Drive, and DiMA+(VAD-Base) tend to have error 4-6 times larger than with a horizon of 1 second. SteerVLA achieves similar performance to Agent-Driver, the current best method on this benchmark that we are aware of.

## A3 LABEL REFINEMENT

Listing 1: BDDX refinement prompt.

```
# Driving Behavior Refinement Prompt

You are an expert in vehicle dynamics and driving behavior analysis. Your
    task is to interpret natural language descriptions of driving
    behavior by analyzing vehicle ego state data (speed and course over
    time). Your response must include two parts:

1. **Ego State Analysis** - a brief explanation of observed speed and
    course trends over time.
2. **Refined Driving Behavior Description** - a more specific version of
    the original description, enhanced with motion extent and driving
    style.

You are an expert in vehicle dynamics and driving behavior analysis. Your
    task is to interpret and refine natural language descriptions of
    driving behavior by analyzing vehicle ego state data (speed and
    course over time) to produce a **precise and nuanced behavior summary
    **. Your output should describe:

1. **Ego State Analysis** - a brief explanation of observed speed and
    course trends over time.
```

```
2. **Refined Driving Behavior Description** – a more specific version of
   the original description, enhanced with a meaningful modifier  _(e.g
   ., **smooth turning**, **wide turn**, **abrupt stop**, **steady lane
   keeping**)_ and a **driving style**, reflecting the driver's attitude
    or intent
   _(e.g., **cautiously**, **normally**, **aggressively**)_

---

## Input Format

**Driving Description:**
INSERT_BEHAVIOR_DESCRIPTION

**Ego Vehicle States:**
INSERT_EGO_STATE_SEQS

These ego states reflect how the vehicle moved during the described
   behavior.

> **Note:**
> - **Course increasing** --> vehicle is turning **right**
> - **Course decreasing** --> vehicle is turning **left**

---

## Output Guidelines

Your response should contain two sections:

### 1. Ego State Analysis

Analyze the speed and course sequence:
- Describe speed patterns: Is the vehicle accelerating, decelerating, or
   maintaining speed?
- Describe course patterns: Is the vehicle turning sharply, smoothly, or
   going straight?
- Mention time duration and total changes in course or speed.
-

### 2. Refined Driving Behavior Description

Produce a single, natural-language sentence that:
- Refines the driving description with motion extent (e.g., *smooth*, *
   sharp*, *wide*, *slight*)
- Adds driving style (e.g., *cautiously*, *normally*, *aggressively*)
- Grounding the refinement in the observable patterns of the ego vehicle
   states

---

## Notes

- The refined description must not exceed **20 words**.
- Use **speed trends** to judge acceleration or deceleration patterns.
- Use **course change patterns** to assess turning sharpness or
   trajectory smoothness.
- If the style cannot be confidently inferred, default to **"normally"**.
- Use **natural, human-readable language**-avoid unnecessary technical
   jargon.

---
## Output Format (REQUIRED)
```

```
Respond **only** with a valid JSON object in the following structure (do
    not include any other text outside the JSON block):

```json
{
  "ego_state_analysis": "<Short paragraph analyzing speed and course
      trends>",
  "refined_description": "<One complete sentence with refined behavior
      and driving style within 20 words>"
}
```,
```

## A4    NUSCENES META-ACTION LABELING

Listing 2: Example Meta Action Labeling Prompt.

```
Prompt 1:
You are an expert in vehicle dynamics and driving behavior analysis. You
    have been provided two frames from a dashcam video from a vehicle,
    with a projected green, yellow, and red trajectory overlaid on the
    first and middle frames of the video of the trajectory that the
    vehicle is in the process of taking. The images are labelled "First
    Frame" and "Middle Frame" at the tops of the images.

Describe:

1. Ego State Analysis:

Analyze the speed and course sequence:
- Describe speed patterns: Is the vehicle accelerating, decelerating, or
    maintaining speed?
- Describe course patterns: Is the vehicle turning sharply, smoothly, or
    going straight?
- Mention time duration and total changes in course or speed.

These ego states reflect how the vehicle moved during the described
    behavior.

> **Note:**
> - **Course increasing**     vehicle is moving **right**
> - **Course decreasing**     vehicle is moving **left**

{ego_states_text}

2. First frame description:
- Describe the lane markings in the first frame image, and the projected
    trajectory's position relative to them at the beginning of the
    trajectory and at the end. Identify any areas on the road with solid
    white or yellow lines.
- Are there road markings, signs, or other structures that indicate that
    the vehicle is at an intersection?
- Which lane does the trajectory begin in, and which lane does the
    trajectory end in?
- Is the red, yellow, and/or green trajectory to the right or left of the
     lane markings?
- Is the cyan circle to the right or left of the lane markings?
- Is the trajectory curving? If so, which way is the trajectory curving?

3. Middle frame description:
- Describe the lane markings in the middle frame image, and the projected
     trajectory's position relative to them at the beginning of the
    trajectory and at the end. Identify any areas on the road with solid
    white or yellow lines.
```

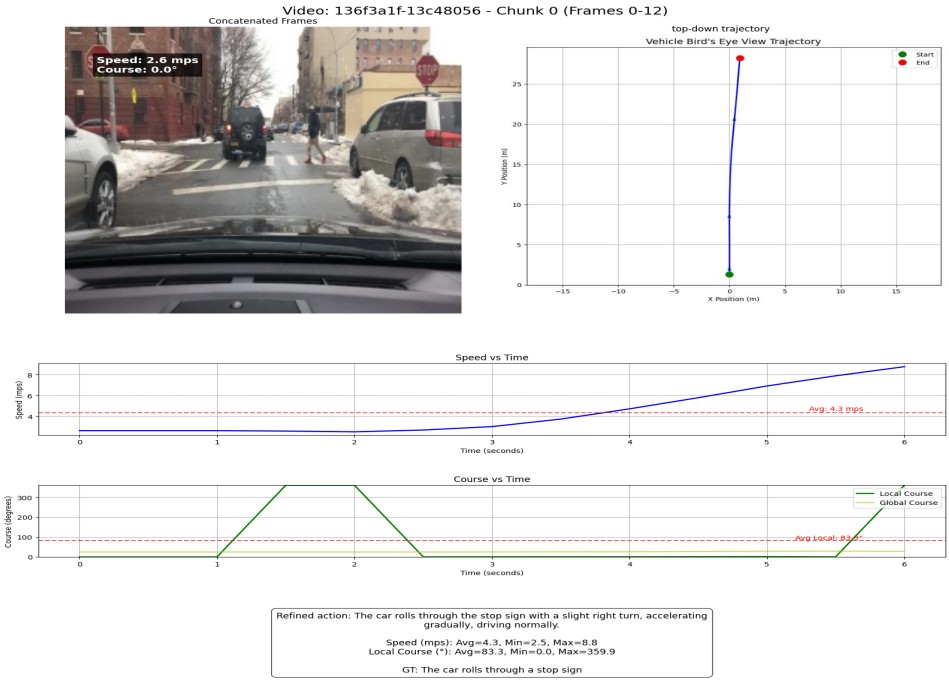

(a) Starting with the label "The car accelerates slowly", we can augment with additional information from the vehicle's states to get the label "The car rolls through the stop sign with a slight right turn, accelerating gradually, driving normally."

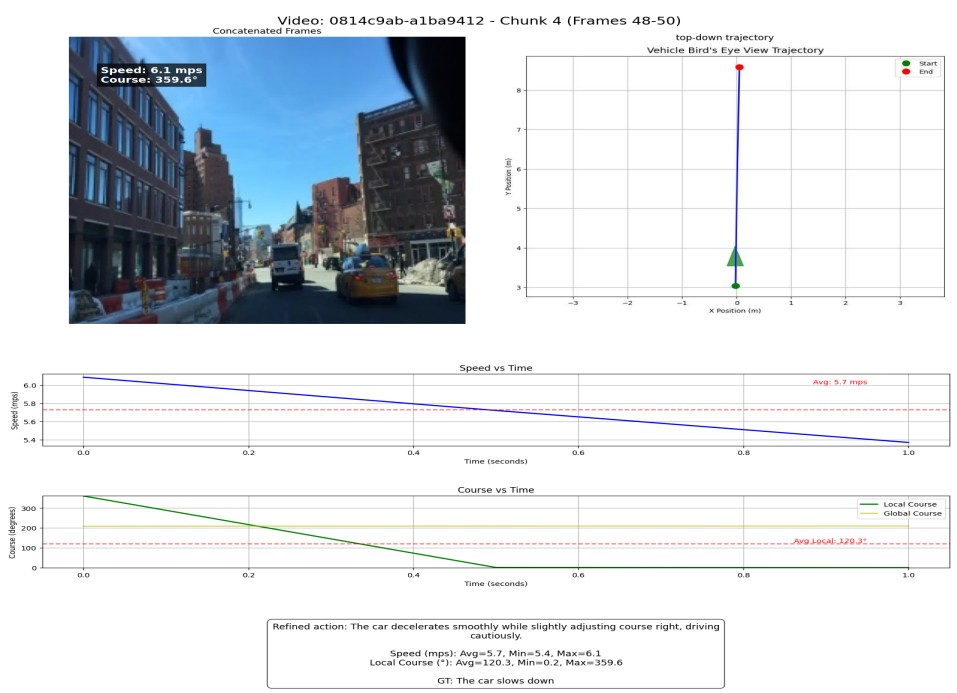

(b) Starting with the label "The car slows down", we can augment with additional information from the vehicle's states to get the label "The car decelerates smoothly while slightly adjusting course right, driving cautiously."

Figure 4: Examples of refining the BDD-X labels to train a more steerable low-level policy.

```
- Are there road markings, signs, or other structures that indicate that
    the vehicle is at an intersection?
- Which lane does the trajectory begin in, and which lane does the
    trajectory end in?
- Is the red, yellow, and/or green trajectory to the right or left of the
    lane markings?
- Is the cyan circle to the right or left of the lane markings?
- Is the trajectory curving? If so, which way is the trajectory curving?

4. Consolidated Analysis:
- Based on your analysis of the first frame image and the middle frame
    image, which lane does the vehicle begin in, and which lane does it
    end in?
- Does this signify a lane change? If so, is the vehicle making a lane
    change to the left, or a lane change to the right?
- Alternatively, is the vehicle at an intersection in either frame? Does
    this signify a turn? Even if the trajectory is curving, consider
    whether the course change is large enough to be a turn, and whether
    the vehicle is simply continuing forward to a parallel road.
- If so, is the vehicle turning to the left, or to the right?

5. Vehicle Action: The action that the vehicle is taking. Is the vehicle
    **turning**, **changing lanes**, or **continuing straight**? If the
    vehicle is turning or changing lanes, is it doing so to the **left**
    or to the **right**? Choose from one of the following discrete
    actions:
- turning left
- turning right
- changing lanes left
- changing lanes right
- continuing straight
- completely stationary
- making a U-Turn

Notes:
- The cyan circle denotes the **end** of the trajectory.
- The trajectory begins at the **bottom** of the image.
- A turn is defined as a full turn at an intersection.
- Otherwise, if the trajectory is simply following a curve in the road,
    describe this as **continuing straight**
- If the trajectory is **continuing straight** through an intersection,
    describe this as **continuing straight**
- If the vehicle has crossed a lane marking, it is most likely making a
    lane change.
- There may be no visible trajectory projected, in which case the vehicle
     is most likely moving very slowly or stationary.
- Identify only the lane markings that are clearly discernible.
- Small course changes of fewer than 4 degrees most likely indicate that
    the vehicle is **continuing forward**.
- Large course changes over 50 degrees likely indicate that the vehicle
    is **turning**.
- Small velocities below 1.0 meters per second likely indicate that the
    vehicle is stationary.

Lane information: {lane_information}

Prompt 2:
# Driving Behavior Refinement Prompt

You are an expert in vehicle dynamics and driving behavior analysis. Your
    task is to interpret and refine natural language descriptions of
    driving behavior by analyzing vehicle ego state data (speed and
    course over time) to produce a **precise and nuanced behavior summary
    **. Your output should describe:
```

```
1. **Ego State Analysis**     a brief explanation of observed speed and
    course trends over time.
2. **Refined Driving Behavior Description**     a more specific version
    of the original description, enhanced with a meaningful modifier _(e.
    g., **smooth turning**, **wide turn**, **abrupt stop**, **steady lane
     keeping**)_ and a **driving style**, reflecting the driver's
    attitude or intent _(e.g., **cautiously**, **normally**, **
    aggressively**)_

---

## Input Format

**Driving Description:**
{driving_description}

**Ego Vehicle States:**
{ego_state_sequence}

These ego states reflect how the vehicle moved during the described
    behavior.

> **Note:**
> - **Course increasing**     vehicle is moving **right**
> - **Course decreasing**     vehicle is moving **left**

---

## Output Guidelines

Your response should contain two sections:

### 1. Ego State Analysis

Analyze the speed and course sequence:
- Describe speed patterns: Is the vehicle accelerating, decelerating, or
    maintaining speed?
- Describe course patterns: Is the vehicle turning sharply, smoothly, or
    going straight?
- Mention time duration and total changes in course or speed.

### 2. Refined Driving Behavior Description

Produce a single, natural-language sentence that:
- Refines the driving description with motion extent (e.g., *smooth*, *
    sharp*, *wide*, *slight*)
- Adds driving style (e.g., *cautiously*, *normally*, *aggressively*)
- Grounding the refinement in the observable patterns of the ego vehicle
    states

---

## Notes

- The refined description must not exceed **20 words**.
- Use **speed trends** to judge accelerationor deceleration patterns.
- Use **course change patterns** to assess turning sharpness or
    trajectory smoothness.
- If the style cannot be confidently inferred, default to **"normally"**.
- Use **natural, human-readable language**  avoid  unnecessary technical
    jargon.
- If the driving descripton is "The vehicle is continuing straight",
    describe any left or right movements as "adjusting left" or "
    adjusting right" respectively. Do not describe this as turning.
```

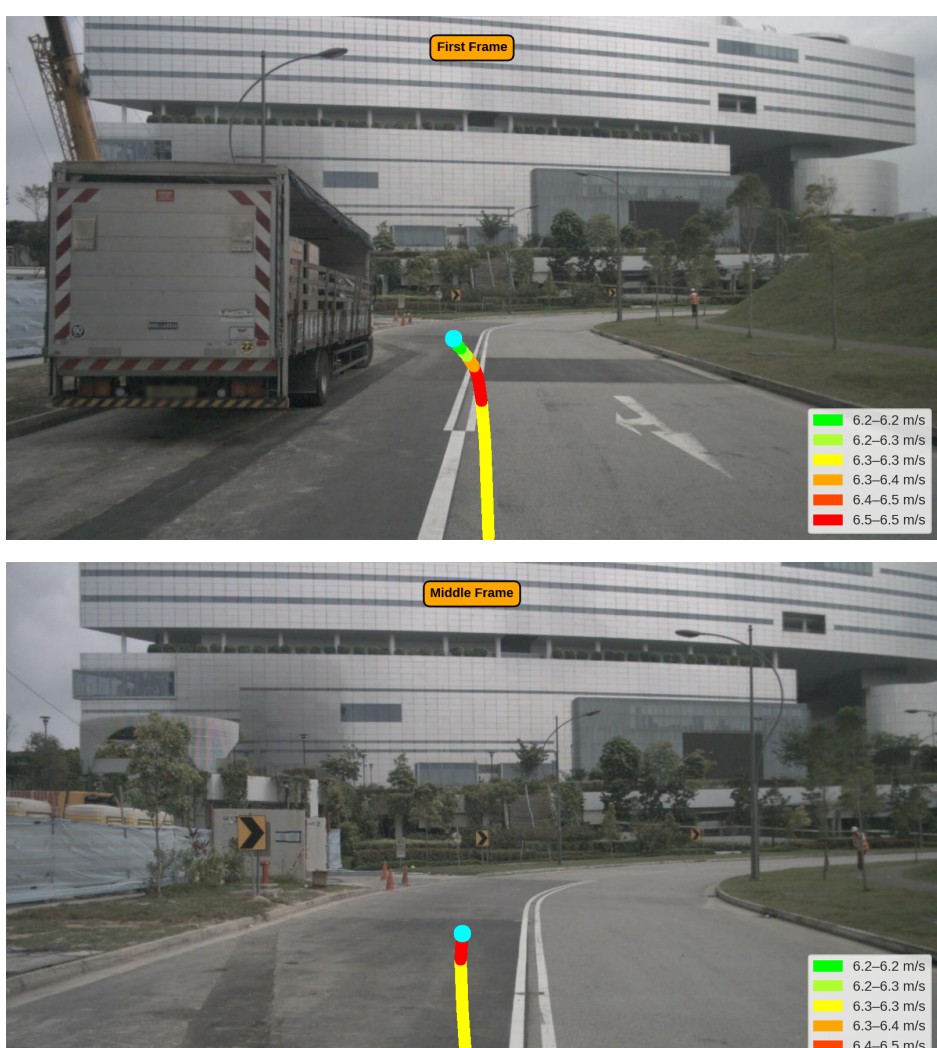

Figure 5: The input images for meta-action labeling. The first-round prompt gives us a simple baseline action ("changing lanes left") and the second-round prompt gives us our refined meta-action ("The vehicle is smoothly changing lanes left normally.")

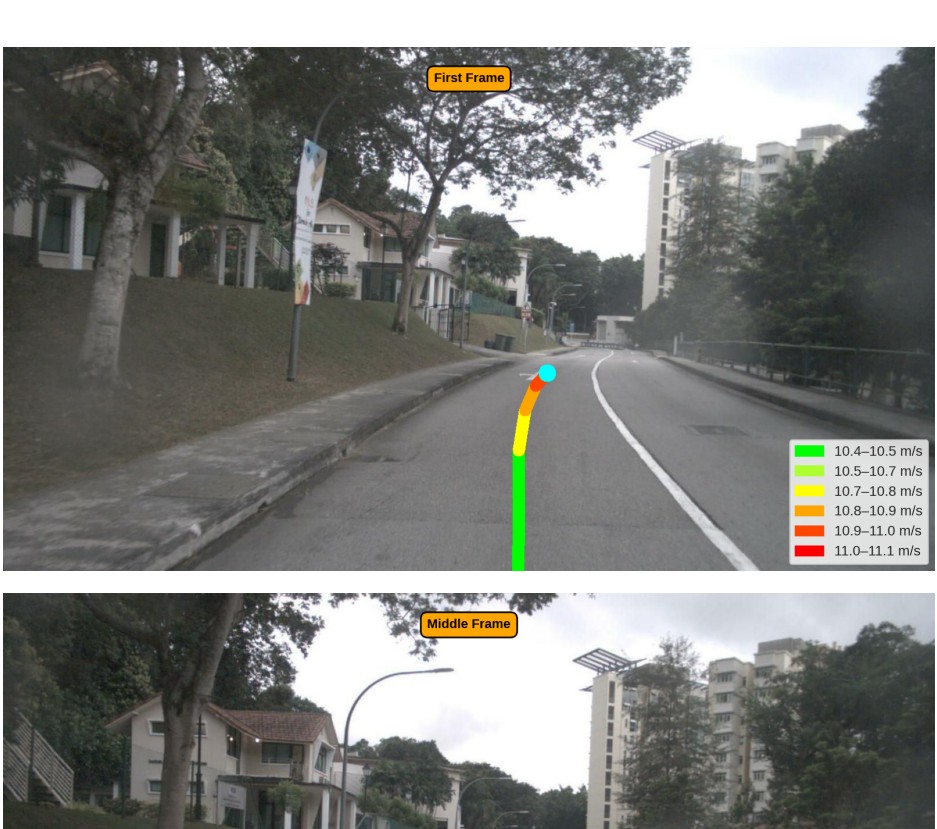

Figure 6: The input images for meta-action labeling. The first-round prompt gives us a simple baseline action ("continuing straight") and the second-round prompt gives us our refined meta-action ("The car normally accelerates, then maintains speed while subtly drifting right.")

