# OpenReview forum: "SteerVLA: Steering Vision-Language-Action Models Toward Effective Long-Tail Driving"
_ICLR.cc/2026/Conference — Submitted to ICLR 2026_

### Official Review · Reviewer_PUZ2 · 2025-10-23

**Soundness:** 2
**Presentation:** 2
**Contribution:** 1
**Rating:** 2
**Confidence:** 3

**Summary:**

The paper “SteerVLA: Steering Vision-Language-Action Models Toward Effective Long-Tail Driving” proposes a hierarchical vision-language-action (VLA) framework intended to improve autonomous driving in long-tail scenarios. The approach involves a high-level vision-language model (VLM) planner that generates meta-actions and reasoning traces, and a low-level controller that executes continuous control actions. The authors also introduce an auto-labeling pipeline to enhance data quality and claim to construct a new Bench2Drive-Hard subset to evaluate performance on challenging scenarios.

While the paper aims to address an important problem—long-tail generalization in autonomous driving—its contributions appear incremental relative to prior work such as DriveVLM and DriveGPT. Furthermore, several methodological and experimental claims are under-specified or insufficiently supported, which limits confidence in the novelty and validity of the results.

**Strengths:**

The focus on long-tail driving and hierarchical perception-action modeling is timely and aligns with ongoing efforts in interpretable and robust autonomous driving.

The modular reasoning-control formulation is conceptually intuitive and potentially extensible to other embodied tasks.

**Weaknesses:**

The weaknesses are listed here:

1. The paper claims that it introduces a hierarchical vision-language-action decomposition, a novel design separating reasoning (language domain) from control (action domain). However, I believe DriveVLM(https://arxiv.org/pdf/2402.12289v4) already proposes a decomposed VLM + action planner system. And it has very similar design in terms of using a VLM to output meta action and a low-level planner outputs trajectory.

2. The data generation pipeline is similar to label generation pipeline proposed from DriveVLM, DriveGPT, and etc. I don't find significant novelty in this step.

3. The author stated that they created a subset of long-tail data from Bench2Drive, called Bench2Drive-Hard, to rigorously benchmark the model’s long-tail driving abilities. However, I didn't find any text describing how this Bench2Drive-Hard dataset is generated. It was not even mentioned out of the introduction paragraph. I am not even sure if the dataset is being used for evaluation.

4. The results in Table1 does not show that SteerVLA has a significant lead over SimLingo. And based on my understanding, SimLingo's system is around 1B parameters and is much smaller compared to the 4B system used by SteerVLA. The marginal improvements may come from the additional model parameters rather than proposed methodology.

**Questions:**

1. What's the major novelty in the data creation step, compared to DriveVLM and DriveGPT?
2. How is your proposed method better than DriveVLM? (even from a conceptual perspective)
3. How is Bench2Drive-Hard generated? And how is it used in this paper?

---

> ### Author Response · Authors · 2025-11-25
> **Response to Reviewer PUZ2 (Part 1/2)**
>
> We are grateful for the reviewer’s suggestions. We address the concerns by clarifying our approach and presenting further supporting experiments.
>
> ## **[Comment 1 & Question 2] “The paper claims that it introduces a hierarchical vision-language-action decomposition...However, I believe DriveVLM already proposes a decomposed VLM + action planner system...**
>
> We thank the reviewers for pointing out additional related work. We included DriveVLM in the revised manuscrip.
>
> The central goal of our work is to understand what a model needs in order to succeed on long-tail driving scenarios. Strong reasoning is crucial for handling rare or unseen situations, and we therefore adopt an explicit hierarchical design with a dedicated high-level planner and low-level policy. This stands in contrast to many prior approaches, such as CoT-based unified models and monolithic LLM/VLA architectures (e.g., DriveGPT-4, DriveVLM), where reasoning and control are fused into a single network, often limiting the ability to maintain both detailed reasoning and reliable control.
>
> To highlight the value of this design choice, the table below (also see Table 2 and Table 4 in revised manuscript) reports Bench2Drive results for SteerVLA trained as a unified model with PaliGemma 3B that directly predicts actions without CoT-style intermediates. Our hierarchical model significantly outperforms this unified variant. In our additional experiments, we will further include a CoT-based version of SteerVLA and expand the ablation to better characterize the model capacity and structural requirements needed for strong reasoning capabilities.
>
> | Method                     | Driving Score | Success Rate (%) |
> |----------------------------|---------------|------------------|
> | SteerVLA (unified)         | 67.66         | 43.64            |
> | SteerVLA (w/ original labels)| 77.65         | 57.73            |
> | **SteerVLA (Ours)**        | **86.81**     | **69.55**        |
>
>
> Furthermore, we find that using refined labels produced by our pipeline in Section 3.2 where future ego trajectories are used to infuse motion extent and style into the meta-actions, substantially improves the high-level planner’s ability to specify precise motions for the low-level policy to execute. This makes the high-level planner more reliable when directing the low-level policy in unfamiliar scenarios, as reflected in the open-loop BDD-X results in Table 5. As shown in the table above, we also report an ablation on the impact of label refinement within the SimLingo dataset under our closed-loop CARLA evaluation setting.
>
> ## **[Comment 2 & Question 1] “The data generation pipeline is similar to label generation pipeline proposed from DriveVLM, DriveGPT, and etc.”**
> Unlike DriveVLM and DriveGPT, our method requires no human annotations and is fully dataset-agnostic. For an unlabeled dataset, we segment each trajectory into 2–5 s chunks and query a VLM using the projected future path, lane context, and ego-state trends to obtain a coarse baseline action (e.g., turn, lane change, go straight). Because this baseline is still limited, we reuse the trajectory information, such as the linear and angular velocity profile, to enrich the label with motion extent and style, producing refined meta-actions like “accelerate with a slight right turn, driving aggressively.”
>
> In addition, we generate reasoning traces through an additional VLM prompt that describes the scene and analyzes surrounding agents. These reasoning traces are learned by the high-level planner as an auxiliary task, but only the refined meta-action is passed to the low-level policy.
>
> ## **[Comment 3 & Question 3] “How is Bench2Drive-Hard generated? And how is it used in this paper?”**
>
> In our original submission, we used Bench2Drive-Hard as a benchmark for our ablation study (please see Section 5.1 and Table 2).
> Bench2Drive-Hard is a subset of Bench2Drive consisting of 10 routes (4, 72, 71, 174, 152, 74, 214, 219, 41, 68). These routes were specifically selected to cover a diverse set of challenging scenarios, such as obstacle avoidance, overtaking and merging, emergency braking, traffic signs, and yielding, allowing us to highlight the reasoning capabilities and limitations of each model variant.
>
> For a more comprehensive evaluation, we perform the ablation study on the entire Bench2Drive benchmark (all 220 routes) and report the corresponding results in the updated table in the response to [Comment 1 & Question 2].

---

> ### Author Response · Authors · 2025-11-25
> **Response to Reviewer PUZ2 (Part 2/2)**
>
> ## **[Comment 4] “The results in Table1 does not show that SteerVLA has a significant lead over SimLingo. And based on my understanding, SimLingo's system is around 1B parameter...”**
> In our original submission, we removed the “force-to-move” heuristic to evaluate the driving policy rather than the full driving system. Since the SimLingo paper reports numbers with this heuristic enabled, we re-enabled it here for a fair comparison. As shown in the table below (also see Table 1 in revised manuscript), SteerVLA achieves the best overall performance on Bench2Drive. While the overall improvement over SimLingo is modest, SteerVLA outperforms SimLingo across all scenario categories in the multi-ability benchmark, particularly in Merging and Give Way, where the gains are more substantial.
>
> In addition, unlike SimLingo, our model is not trained with Action Dreaming data. Training with Action Dreaming is impractical in real-world settings because it requires privileged simulator states. The “safe” flag in Action Dreaming depends on internal simulator information (ground-truth positions, geometry, and collision checks) that cannot be obtained from sensors alone. For context, the SimLingo paper reports that the variant without Action Dreaming achieves only a 84.41 driving score and 64.85 success rate.
>
> | Method               | DS ↑    | SR (%) ↑ | --- | **Ability (%) ↑** |        |        |        |        |        |
> |----------------------|---------|-----------|-----|---------------------|--------|--------|--------|--------|--------|
> |                      |         |           |     | Merging | Overtaking | Emergency Brake | Give Way | Traffic Sign | Mean |\
> | DriveMoE             | 74.22   | 48.64     |     | 34.67   | 40.00      | 65.45            | 40.00    | 59.44         | 47.91 |
> | ORION                | 77.74   | 54.62     |     | 25.00   | **71.11**  | **78.33**        | 30.00    | 69.15     | 54.72 |
> | AutoVLA              | 78.84   | 57.73     |     | -       | -          | -                | -        | -             | -     |
> | SimLingo             | 85.94   | 66.82     |     | 57.50   | 60.00      | 76.67            | 50.00    | 73.16         | 63.46 |
> | **SteerVLA (Ours)**  | **86.81** | **69.55** |     | **66.25** | 61.36  | 76.67           | **80.00** | **76.32**        | **72.12** |
>
>
> It is worth noting that, in the table above, DriveMoE also uses PaliGemma-3B, which is the same backbone as ours.
>
> We added an ablation study on different base models for the high-level planner in Section 5.1 and Table 2 in revised manuscript.
>
> We observe that using the InternVL-2 as the base model for the high-level planner hurts performance, while outperforming the unified models. InternVL-2 is trained a multi-modal reasoning model, focusing primarily on vision and image understanding, while Gemma3 is a powerful, generalist reasoning model. It is possible that InternVl-2 cannot handle the more complex language of the refined labels we train on, meaning that inherent strong reasoning capabilities in the base model is necessary for SteerVLA.
>
>
> Thank you for the thoughtful review. Please let us know if our responses address your concerns, and we would be happy to clarify anything further.

---

### Official Review · Reviewer_wN4f · 2025-10-27

**Soundness:** 2
**Presentation:** 3
**Contribution:** 2
**Rating:** 4
**Confidence:** 5

**Summary:**

This paper proposes a two stage hierarchical architecture for VLA for autonomous driving. The higher level is done by a VLM, fine-tuned on driving related visual question answering task. This produces both meta actions and reasoning traces about the scene and action. The lower level planner takes the meta action along with scene information and generates the trajectory. The authors present competitive performance with SOTA methods. There is also a pseudo label generation presented, that can be used for data augmentation in any VLA related tasks.

**Strengths:**

1. By splitting the driving task into two stages makes it flexible, adaptable to various scenarios. Even though hierarchical design is not new, two stage VLA design is quite new. This can potentially improve generalizability, especially helping with corner cases using the higher level reasoning for better overall performance.

2. Refinement step can be seen as a generic data augmentation method that can be isolated out of this work and applicable to any other labeling use cases.

**Weaknesses:**

1. Hierarchical design also introduces some complexities that the paper does not address. Like, choosing inference frequencies of the two stages or dealing with synchronization issues between the two systems are  difficult problems that should have been mentioned.

2. The evaluation logic for the refinement step is questionable. As, speed and turning angle are what is used for refinement and also being used for measurement, this could cause data leakage issue. Ideally, this should be evaluated on a different dataset or with non-overlapping chunks using careful masking of the future state information.

**Questions:**

1. The decision to remove 'force-to-move' heuristic from Similingo is not justified well. It is important to show full end-to-end system performance. It should have been reported and if possible be included into this method for a fair comparison.

2. The baseline methods mostly use older VLM models like Intern VL2 etc. It is not clear if the performance jump is due to better VLM or from better algorithm design. An ablation study, showing the full system performance using the same VLM as the baseline methods would be convincing.

3. As all the closed loop evaluations are done in simulation, there should be a mention of sim-to-real gaps and some possible ways to address them.

---

> ### Author Response · Authors · 2025-11-25
> **Response to Reviewer wN4f (Part 1/2)**
>
> We appreciate the reviewer’s helpful comments. Below, we refine our explanations and provide new experiments to better support our statements.
>
> ## **[Comment 1] “Hierarchical design also introduces some complexities that the paper does not address...”**
>
> This is a very important question. Hierarchical systems can introduce additional complexity. However, in our design these considerations are simple to handle. For inference efficiency, we run the high-level planner at a lower frequency and the low-level policy at a higher one; in practice, the planner operates at 5 Hz while the low-level policy runs at 20 Hz. This setting provides stable long-horizon reasoning from the planner and fast reactive control from the policy. (See Section 4.1 in revised manuscript)
>
> Synchronization is also straightforward because the two components communicate through a single meta-action string. The low-level policy always reads the most recent meta-action from the planner, so no extra alignment or coordination mechanisms are required. This clean language interface ensures consistent behavior without the typical synchronization challenges of hierarchical control systems.
>
> ## **[Comment 2] “The evaluation logic for the refinement step is questionable. As, speed and turning angle are what is used for refinement and also being used for measurement, this could cause data leakage issue...”**
>
> We would like to clarify that using speed and turning angle during data refinement does not introduce data leakage. The refined labels are used **only as supervision signals during training**, not as inputs to the policy. During inference, the model does not receive any future kinematic information; it only takes current (and historical) raw sensory observations and the routing command. The model then predicts a future trajectory, which can be represented as future speeds and turning angles, but these quantities are **generated by the model**, not provided to it.
>
> In other words, the refinement step uses speed and turning angle offline to produce cleaner training labels, but the policy never has access to future states at test time. Therefore, there is no leakage from refinement features into the inference inputs, and the evaluation remains valid.
>
> ## **[Question 1] “The decision to remove 'force-to-move' heuristic from Similingo is not justified well...”**
>
> In our original submission, we removed the “force-to-move” heuristic to evaluate the driving policy rather than the full driving system. Since the SimLingo paper reports numbers with this heuristic enabled, we reenabled it here for a fair comparison. As shown in the table below (also see Table 1 in the revised manuscript), SteerVLA achieves the best overall performance on Bench2Drive. While the overall improvement over SimLingo is modest, SteerVLA outperforms SimLingo across all scenario categories in the multi-ability benchmark, particularly in Merging and Give Way, where the gains are more substantial.
>
> | Method               | DS ↑    | SR (%) ↑ | --- | **Ability (%) ↑** |        |        |        |        |        |
> |----------------------|---------|-----------|-----|---------------------|--------|--------|--------|--------|--------|
> |                      |         |           |     | Merging | Overtaking | Emergency Brake | Give Way | Traffic Sign | Mean |\
> | DriveMoE             | 74.22   | 48.64     |     | 34.67   | 40.00      | 65.45            | 40.00    | 59.44         | 47.91 |
> | ORION                | 77.74   | 54.62     |     | 25.00   | **71.11**  | **78.33**        | 30.00    | 69.15     | 54.72 |
> | AutoVLA              | 78.84   | 57.73     |     | -       | -          | -                | -        | -             | -     |
> | SimLingo             | 85.94   | 66.82     |     | 57.50   | 60.00      | 76.67            | 50.00    | 73.16         | 63.46 |
> | **SteerVLA (Ours)**  | **86.81** | **69.55** |     | **66.25** | 61.36  | 76.67           | **80.00** | **76.32**        | **72.12** |

---

> ### Author Response · Authors · 2025-11-25
> **Response to Reviewer wN4f (Part 2/2)**
>
> ## **[Question 2] “The baseline methods mostly use older VLM models like Intern VL2 etc...An ablation study, showing the full system performance using the same VLM as the baseline methods would be convincing.”**
>
> We added ablation study on different base models for the high-level planner in Section 5.1 and Table 2 in revised manuscript.
> We observe that using the InternVL-2 as the base model for the high-level planner hurts performance, while outperforming the unified models. InternVL-2 is trained a multi-modal reasoning model, focusing primarily on vision and image understanding, while Gemma3 is a powerful, generalist reasoning model. It is possible that InternVl-2 cannot handle the more complex language of the refined labels we train on, meaning that inherent strong reasoning capabilities in the base model is necessary for SteerVLA.
>
> ## **[Question 3] “As all the closed loop evaluations are done in simulation, there should be a mention of sim-to-real gaps and some possible ways to address them.”**
>
> We include open-loop results on the nuScenes planning benchmark to evaluate SteerVLA’s ability to generalize to real-world data (see Appendix A2 and Table 6 in revised manuscript). Following standard protocol, we predict 1, 2, and 3-second future trajectories at 2 Hz and report the trajectory-level L2 error on the nuScenes validation set. The results are shown below:
>
> |Method|1s|2s| 3s| Avg.|
> |--|--|---|--|--|
> |PARA-Drive [1]|0.26|0.59|1.12|0.66|
> | TOKEN [2]| 0.36 | 0.70 | 1.46 | 0.81 |
> | DiMA+(VAD-Base) [3]| 0.18 | 0.48 | 1.01 | 0.56 |
> | **SteerVLA (Ours)** | **0.18** | **0.39** | **0.63** | **0.40** |
>
> [1] PARADrive, PARA-Drive: Parallelized Architecture for Real-time Autonomous Driving, CVPR 2024.
>
> [2] TOKEN:Tokenize the World into Object-level Knowledge to Address Long-tail Events in Autonomous Driving, CoRL 2024.
>
> [3] DiMA, Distilling Multi-modal Large Language Models for Autonomous Driving, CVPR 2025.
>
> SteerVLA achieves the best performance across all horizons and the lowest average L2 error.
>
> Besides, the SimLingo dataset also spans diverse weather, lighting, and town layouts, helping the low-level policy adapt to visual variations.
>
> However, there remain sim-to-real gaps. Co-training or fine-tuning on real datasets such as nuScenes and BDD-X will be important for handling real sensor noise and long-tail cases not present in CARLA. Another gap comes from the behavior of other agents where CARLA’s NPCs are rule-based and far less diverse than real human drivers. Using data-driven traffic models or incorporating real driving data can further narrow this gap.
>
> Thank you for the thoughtful review. Please let us know if our responses address your concerns, and we would be happy to clarify anything further.

---

### Official Review · Reviewer_83of · 2025-11-01

**Soundness:** 3
**Presentation:** 3
**Contribution:** 2
**Rating:** 4
**Confidence:** 5

**Summary:**

This paper proposes a VLA method named SteerVLA, which focuses on addressing the long-tail driving problem. Specifically, it consists of two stages: a high-level VLM planner that performs semantic and common-sense reasoning to analyze driving scenarios, and a low-level VLA actor that generates precise control actions based on those instructions.

**Strengths:**

1. The motivation is strong, long-tail cases are indeed an important issue in autonomous driving.

2. The experimental results look promising.

**Weaknesses:**

1. Using VLMs to generate reasoning-based labels is not novel. For example, VLM-AD [1] adopts a similar strategy by automatically generating reasoning information through various designed prompts or questions. It would be better to include such related works and discuss how this method differs from or improves upon them.

[1] Y. Xu et al., “VLM-AD: End-to-End Autonomous Driving through Vision-Language Model Supervision,” Proceedings of the 9th Annual Conference on Robot Learning (CoRL), 2025


2. Some explanations are unclear. For example, how are meta-actions generated? How are the language–action labels created? Where does the reasoning originate?

3. As mentioned in the introduction, this work focuses on long-tail cases. However, more experiments could be included to verify this claim. For instance, are there specific long-tail scenarios where the proposed method outperforms existing ones? What do the metrics look like under those corner cases?

**Questions:**

1. How is the quality of the generated labels evaluated? In other works, questionnaires or human evaluations are often used to ensure quality. If the quality is poor, since VLMs cannot guarantee 100% correctness in understanding driving scenarios, especially long-tail cases, does this not affect the reliability of the proposed method?

2. How are the meta-action labels generated? As mentioned in Line 213, it is unclear how these meta-actions are defined. Is there a predefined list of meta-actions? Given the focus on long-tail cases, how is the coverage of possible actions ensured?

3. In Section 3.2, if fine-grained labels are derived from the original ones, does that mean no additional information is introduced? If so, how are new reasoning traces generated?

---

> ### Author Response · Authors · 2025-11-25
> **Response to Reviewer 83of (Part 1/2)**
>
> We thank the reviewer for the constructive and insightful feedback. In this response, we clarify our method and contributions and add experiments that strengthen our claims.
>
> ## **[Comment 1] “Using VLMs to generate reasoning-based labels is not novel. For example, VLM-AD [1] adopts a similar strategy…”**
>
> We thank the reviewer for the additional related work!
>
> We differ from VLM-AD in how text prediction is used. VLM-AD uses text prediction only as an auxiliary task and does not condition its policy on generated text. In our framework, the reasoning trace is likewise an auxiliary objective, but the low-level policy is explicitly conditioned on the predicted meta-actions, which provide structured, high-level guidance to trajectory prediction. Moreover, unlike VLM-AD, our method leverages the internet-scale pretraining of existing VLMs, which we find crucial for generalizing to long-tail driving scenarios.
>
> ## **[Comment 2] “Some explanations are unclear. For example, how are meta-actions generated? How are the language–action labels created? Where does the reasoning originate?”**
>
> Meta-actions are generated through a two-stage VLM labeling pipeline. We first segment each trajectory into 2–5 s chunks and query a VLM with the projected future path, lane context, and ego-state trends to obtain a baseline action (e.g., turn, lane change, go straight). We then perform a refinement step, where the VLM incorporates the vehicle’s speed and course profiles to produce a more fine-grained description, including motion extent and style. The final refined label is used as the meta-action. Language–action labels are therefore created entirely through this VLM-based two-stage querying process. (See Section 3.2)
>
> Reasoning traces originate from an additional VLM prompt that describes the scene and analyzes the motion of other agents. These traces are predicted by the high-level planner during training, but only the meta-action is used to condition the low-level policy. (See Section 3.2)
>
> ## **[Comment 3] “As mentioned in the introduction, this work focuses on long-tail cases...are there specific long-tail scenarios where the proposed method outperforms existing ones?...”**
>
> We evaluate our policy on the multi-ability benchmark in Bench2Drive, which specifically targets long-tail behaviors. SteerVLA achieves the highest mean long-tail ability score (72.12%), outperforming all baselines by a clear margin. Notably, it shows substantial gains in challenging scenarios such as “Give Way” and “Merging”. (See Section 4.2 and Table 1)
>
> | Method             | Merging | Overtaking | Emergency Brake | Give Way | Traffic Sign | Mean   |
> |--------------------|---------|------------|------------------|----------|--------------|--------|
> | DriveMoE           | 34.67   | 40.00      | 65.45            | 40.00    | 59.44        | 47.91 |
> | ORION              | 25.00   | **71.11**  | **78.33**        | 30.00    | 69.15    | 54.72 |
> | AutoVLA            | -       | -          | -                | -        | -            | -     |
> | SimLingo           | 57.50   | 60.00      | 76.67            | 50.00    | 73.16        | 63.46 |
> | **SteerVLA (Ours)**| **66.25** | 61.36    | 76.67            | **80.00**| **76.32**        | **72.12** |
>
> ## **[Question 1.1] “How is the quality of the generated labels evaluated?”**
>
> To assess the benefits of our refined labels, we compare models trained on the original SimLingo labels with those trained on our refined labels. As the results shown in the table below, the refined labels yield significantly better closed-loop performance on Bench2Drive. We also observe qualitative improvements in scenarios that require more fine-grained language instructions. For instance, when the car turns right to merge, it must quickly match the speed of oncoming traffic. By incorporating motion extent into the meta-action during data refinement, the vehicle accelerates appropriately and merges smoothly without collisions. (See Table 4)
>
> | Method                     | Driving Score | Success Rate (%) |
> |----------------------------|---------------|------------------|
> | SteerVLA w/ original labels| 77.65         | 57.73            |
> | **SteerVLA (Ours)**        | **86.81**     | **69.55**        |

---

> ### Author Response · Authors · 2025-11-25
> **Response to Reviewer 83of (Part 2/2)**
>
> ## **[Question 1.2] “In other works, questionnaires or human evaluations are often used to ensure quality.”**
>
> We also do a human evaluation on the labels generated by our data-generation pipeline on the NuScenes dataset. Since NuScenes is unlabeled, our pipeline generates all labels from scratch. We randomly sample 20 examples from three categories and manually evaluate their correctness. (See Table 3 and Section 5.2)
>
> | Category                | Accuracy (%) |
> |-------------------------|--------------|
> | Accelerating/Decelerating | 96           |
> | Turning                 | 84           |
> | Lane Changing           | 68           |
>
> These results show that our pipeline generates reliable labels for clear motion-extent behaviors (e.g., acceleration/deceleration), while performance drops on more subtle and visually ambiguous actions like lane changes. This is expected, as these categories demand finer spatial and temporal cues. Overall, the accuracies demonstrate that our pipeline produces sufficiently high-quality labels even on an unlabeled dataset like NuScenes.
>
> ## **[Question 1.3] “If the quality is poor, ..., does this not affect the reliability of the proposed method?”**
>
> Our low-level policy is trained only on safe trajectories, so it can often compensate for noisy or partially incorrect meta-actions from the high-level planner. However, the system is not fully reliable when the meta-action is significantly wrong. In rare long-tail cases, the planner may hallucinate an unsafe instruction, for instance, mistaking a red light for green in fog and outputting “accelerate”, and the low-level policy may still follow it. This reflects the inherent limitations of VLMs in complex scenarios. In practice, these failures can be mitigated with lightweight deployment-time safety checks (e.g., rule-based or traffic-light sanity filters) to prevent incorrect meta-actions from propagating to control.
>
> ## **[Question 2] “How are the meta-action labels generated? ...Given the focus on long-tail cases, how is the coverage of possible actions ensured?”**
>
> We do not use a fixed list of meta-actions. All meta-actions are generated automatically from trajectory chunks using a two-stage VLM process described in the response to [Comment 3]. Thus, meta-actions are **free-form natural language**, not predefined categories.
>
> Coverage of long-tail cases is achieved because the labels are generated directly from hindsight trajectories: any rare behavior in the driving data results in a corresponding fine-grained meta-action.
>
> ## **[Question 3] “In Section 3.2, if fine-grained labels are derived from the original ones, does that mean no additional information is introduced? If so, how are new reasoning traces generated?”**
>
> Although some fine-grained labels refine existing ones, they do introduce additional information because the refinement step uses ego state signals and future trajectory projections that might not be encoded in the original labels. The VLM incorporates this extra motion information (speed trends, turning curvature, acceleration patterns) to produce richer meta-actions that go beyond the coarse annotations provided in the dataset.
>
> The reasoning traces are generated independently of the original labels. They are produced by prompting the VLM with the full driving clip, asking it to describe the scene and analyze the behavior of surrounding agents. The reasoning traces come from a separate VLM query that uses visual and motion context, not from the dataset’s text. In this way, both the refined meta-actions and the reasoning traces introduce new information extracted from the underlying trajectory and scene, rather than simply rephrasing the original labels.
>
>
> Thank you for the thoughtful review. Please let us know if our responses address your concerns, and we would be happy to clarify anything further.

---

### Official Review · Reviewer_dqkw · 2025-11-01

**Soundness:** 2
**Presentation:** 2
**Contribution:** 2
**Rating:** 2
**Confidence:** 4

**Summary:**

The paper proposes SteerVLA, to address the critical or long-tail scenarios of autonomous driving, where the proposed SteerVLA contains a hierarchical vision–language–action (VLA) framework aimed at improving complex reasoning and adaptability. SteerVLA consists of

- High-level VLM planner: Performs semantic reasoning and outputs meta-actions (language-based driving instructions) and reasoning traces.
- Low-level VLA actor: Converts meta-actions into precise control actions such as waypoints.
To train these components, the authors introduce an auto-labeling pipeline that generates fine-grained language labels and reasoning traces from driving data.

**Strengths:**

- SteerrVLA contains separate reasoning and control, reducing knowledge collapse and improving generalization.
- SteerVLA's data augmentation pipeline generates nuanced language labels and reasoning traces, improving steerability and language following.
- SteerVLA's Interpretability benefits meta-actions and reasoning traces provide transparency in decision-making.

**Weaknesses:**

- SteerVLA's claims they provided open-loop evaluations, but the Table is closed-loop evaluations on Bench2Drive, Table 2 is ablation study, and Table 3 is VLA policy on trajectory prediction. But there is open-loop evaluations are missing. As the authors claim the SteerVLA improves performance on long-tail scenarios. SteerVLA should compare with TOKEN, PARADrive and DiMA where they performed evaluation on long-tail scenarios on nuScenes Dataset. Or Authors should compare with AutoVLA on NAV-Hard test dataset to see how SteerVLA performs on critical autonomous driving cases.
- The performance improvements of SteerVLA is minor, SteerVLA achieves a Driving score of 81.99 on Bench2Drive dataset, but SimLingo-Baseline achieves 85.94 driving score using 0.5 Billion model.
- SteerVLA  claims  proposed data generation pipeline that can generate refined language labels, their experiments to show this claim are inadequate. The authors only one visualization example.

- TOKEN:Tokenize the World into Object-level Knowledge to Address Long-tail Events in Autonomous Driving, CoRL 2024.
- PARADrive, PARA-Drive: Parallelized Architecture for Real-time Autonomous Driving, CVPR 2024.
- DiMA, Distilling Multi-modal Large Language Models for Autonomous Driving, CVPR 2025.
- AutoVLA: A Vision-Language-Action Model for End-to-End Autonomous Driving with Adaptive Reasoning and Reinforcement Fine-Tuning, CVPR 2025.

**Questions:**

- How robust is the auto-labeling pipeline of SteerVLA to domain shifts (e.g., different countries, weather, or sensor setups)?
- What are the failure cases observed during evaluation, and how severe are they in terms of safety?
- How does SteerVLA handle contradictory or ambiguous meta-actions in real-time traffic?
- Can the hierarchical approach introduce latency issues in high-speed scenarios? Any benchmarks on inference time?

---

> ### Author Response · Authors · 2025-11-25
> **Response to Reviewer dqkw (Part 1/2)**
>
> We sincerely thank the reviewer for the constructive and insightful feedback. In this response, we clarify our method and contributions and present additional experiments to further substantiate our claims.
>
> ## **[Comment 1] “Open-loop evaluations are missing… SteerVLA should compare with TOKEN, PARADrive and DiMA where they performed evaluation on long-tail scenarios on nuScenes Dataset.”**
>
> Please see Appendix A1 and Table 5 in our revised manuscript, which includes an open-loop evaluation study on the BDD-X dataset. For convenience, we summarize the study here: we compare against DriveGPT4 in terms of RMSE on speed and turning angle, showing that both our architecture and the label refinement pipeline yield improved performance.
>
> We also thank the reviewer for pointing out additional related works. We have **added comparisons to TOKEN, PARA-Drive, and DiMA** on the nuScenes validation set. Following their protocol, we predict future trajectories at a 2 Hz sampling rate over 1/2/3 seconds and report the trajectory-level L2 error. The results are shown below:
>
> |Method|1s|2s| 3s| Avg.|
> |--|--|---|--|--|
> |PARA-Drive|0.26|0.59|1.12|0.66|
> | TOKEN| 0.36 | 0.70 | 1.46 | 0.81 |
> | DiMA+(VAD-Base)| 0.18 | 0.48 | 1.01 | 0.56 |
> | **SteerVLA (Ours)** | **0.18** | **0.39** | **0.63** | **0.40** |
>
> We have added this result in Appendix A2 and Table 6 in the revised manuscript.
>
> We would also like to note that the closed-loop evaluation on Bench2Drive in our study provides a much more meaningful assessment than open-loop metrics, as closed-loop evaluation captures control dynamics, recovery behavior, and holistic policy performance. In contrast, open-loop evaluation only measures how closely the model matches expert trajectories on fixed data. We hope these new results further validate our effectiveness in the open-loop setting while encouraging the reviewer to place greater emphasis on the closed-loop evaluation.
>
> ## **[Comment 2.1] “The performance improvements of SteerVLA are minor.”**
>
> In our original submission, we removed the “force-to-move” heuristic to evaluate the driving policy rather than the full driving system. Since the SimLingo paper reports numbers with this heuristic enabled, we reenabled it here for a fair comparison. As shown in the table (also see Table 1 in revised manuscript), SteerVLA achieves the best overall performance on Bench2Drive. While the overall improvement over SimLingo is modest, SteerVLA outperforms SimLingo across all scenario categories in the multi-ability benchmark, particularly in Merging and Give Way, where the gains are more substantial.
>
> In addition, unlike SimLingo, our model is not trained with Action Dreaming data. Training with Action Dreaming is impractical in real-world settings because it requires privileged simulator states. The “safe” flag in Action Dreaming depends on internal simulator information (ground-truth positions, geometry, and collision checks) that cannot be obtained from sensors alone. For context, the SimLingo paper reports that the variant without Action Dreaming achieves only a 84.41 driving score and 64.85 success rate.
>
> | Method               | DS ↑    | SR (%) ↑ | --- | **Ability (%) ↑** |        |        |        |        |        |
> |----------------------|---------|-----------|-----|---------------------|--------|--------|--------|--------|--------|
> |                      |         |           |     | Merging | Overtaking | Emergency Brake | Give Way | Traffic Sign | Mean |\
> | DriveMoE             | 74.22   | 48.64     |     | 34.67   | 40.00      | 65.45            | 40.00    | 59.44         | 47.91 |
> | ORION                | 77.74   | 54.62     |     | 25.00   | **71.11**  | **78.33**        | 30.00    | 69.15     | 54.72 |
> | AutoVLA              | 78.84   | 57.73     |     | -       | -          | -                | -        | -             | -     |
> | SimLingo             | 85.94   | 66.82     |     | 57.50   | 60.00      | 76.67            | 50.00    | 73.16         | 63.46 |
> | **SteerVLA (Ours)**  | **86.81** | **69.55** |     | **66.25** | 61.36  | 76.67           | **80.00** | **76.32**        | **72.12** |
>
> ## **[Comment 2.2] “SimLingo-Baseline achieves 85.94 driving score using 0.5 Billion model.”**
>
> SteerVLA’s improvements are not due to model size: DriveMoE uses the same PaliGemma-3B backbone yet performs worse.
>
> We added ablation study on different base models for the high-level planner in Section 5.1 and Table 2 in revised manuscript.
>
> We observe that using the InternVL-2 as the base model for the high-level planner hurts performance, while outperforming the unified models. InternVL-2 is trained a multi-modal reasoning model, focusing primarily on vision and image understanding, while Gemma3 is a powerful, generalist reasoning model. It is possible that InternVl-2 cannot handle the more complex language of the refined labels we train on, meaning that inherent strong reasoning capabilities in the base model
> is necessary for SteerVLA.

---

> ### Author Response · Authors · 2025-11-25
> **Response to Reviewer dqkw (Part 2/2)**
>
> ## **[Comment 3] “experiments to show this claim (‘data generation pipeline that can generate refined language labels’) are inadequate. ”**
>
> To assess the importance of our data-generation pipeline, we compare models trained on the original SimLingo labels with those trained on our refined labels. As the results shown in the table below (also see **Table 4** in our revised manuscript), the refined labels yield significantly better closed-loop performance on Bench2Drive. We also observe qualitative improvements in scenarios that require more fine-grained language instructions. For instance, when the car turns right to merge, it must quickly match the speed of oncoming traffic. By incorporating motion extent into the meta-action during data refinement, the vehicle accelerates appropriately and merges smoothly without collisions.
>
> | Method                     | Driving Score | Success Rate (%) |
> |----------------------------|---------------|------------------|
> | SteerVLA w/ original labels| 77.65         | 57.73            |
> | **SteerVLA (Ours)**        | **86.81**     | **69.55**        |
>
> We hope the new results provide stronger support to our claim.
>
> ## **[Question 1] “How robust is the auto-labeling pipeline of SteerVLA to domain shifts (e.g., different countries, weather, or sensor setups)?”**
>
> The 220 routes in the Bench2Drive benchmark span 23 weather conditions (sunny, foggy, rainy, etc.) and 12 towns (urban, village, university, etc.), and include diverse long-tail scenarios such as yielding to jaywalkers, navigating around construction sites, and handling ambiguous right-of-way interactions. In the table provided in our last answer, the ablation study compares refined labels to the original labels on Bench2Drive, which validates the robustness of our auto-labeling pipeline.
>
> We added this description of Bench2Drive in **Section 4.1** of the revised manuscript.
>
> ## **[Question 2] “What are the failure cases observed during evaluation, and how severe are they in terms of safety?”**
>
> One main failure case is that the model can incorrectly interpret a parked vehicle as moving, which may lead to unsafe acceleration. This arises from occasional hallucinations in our VLM-based labeling pipeline that assign incorrect motion states to surrounding vehicles. Another major failure mode occurs under degraded weather conditions, where the model may misdetect critical scene elements, for instance, incorrectly classifying traffic-light states in foggy conditions. These are important safety-critical issues that require more careful engineering, such as improving temporal consistency checks and using stronger multi-frame validation to reduce label noise.
>
> We added this in Section 5 of the revised manuscript.
>
> ## **[Question 3] “How does SteerVLA handle contradictory or ambiguous meta-actions in real-time traffic?”**
>
> When the high-level planner outputs a contradictory or ambiguous meta-action, the low-level policy often still produces a safe trajectory because it is trained only on safe data (we filter out trajectories with low driving scores and unsafe behavior from the training data). However, the system can still fail. For example, in foggy weather the high-level planner may hallucinate a green light and output “accelerate” at a red-light junction, and the low-level policy may follow this instruction. This can be solved by some practical tricks like basic rule checks when deploying the policy in the real world.
>
> ## **[Question 4] “Can the hierarchical approach introduce latency issues in high-speed scenarios? Any benchmarks on inference time?”**
>
> The high-level planner has an inference latency of 0.55 s per forward pass on an A100 GPU, while the low-level policy has a latency of 0.69 s on an A100 GPU. During policy deployment, the high-level planner operates at 5 Hz and the low-level policy runs at 20 Hz. Given that the high-level planner is called at a much lower frequency, its amortized per-step cost is substantially smaller in practice. Although we have not dedicated significant effort to optimizing inference, we note that the runtime could be further reduced using lightweight caching mechanisms, for instance, applying KV-cache–based caching to avoid recomputing repeated context. (See Section 4.1 in revised manuscript)
>
> Thank you for the thoughtful review. Please let us know if our responses address your concerns, and we would be happy to clarify anything further.

---

### Author Response · Authors · 2025-12-03
**General Response**

We thank all reviewers for their insightful comments and questions. Below, we address the main concerns raised.

## Main Concerns Addressed

### Performance improvements compared with baseline methods (**dqkw**, **wN4f**, **PUZ2**)
With the force-to-move heuristic re-enabled for a fair comparison, our policy achieves the best overall performance on Bench2Drive, consistently surpassing SimLingo across all ability categories, particularly in long-tail scenarios such as Merging and Give Way.
(See response to **Comment 2.1** from reviewer **dqkw** and **Table 1** in the revised manuscript)

### Clarification of contributions compared to prior work (**83of**, **PUZ2**)
We explicitly compare our hierarchical design to methods such as DriveGPT-4 and DriveVLM, and clarify how our auto-labeling pipeline differs from related work including VLM-AD and DriveVLM.
(See response to **Comment 1** from reviewer **83of**, and responses to **Comments 1** and **2** from reviewer **PUZ2**)

### Ablation studies (**dqkw**, **83of**, **wN4f**, **PUZ2**)
- **Ablation of the high-level policy** to evaluate the impact of different base models and the importance of our hierarchical architecture
  (See response to **Comment 2.2** from reviewer **dqkw**, response to **Comment 1** from reviewer **PUZ2**, **Table 2** and **Section 5.1**)

- **Ablation of the meta-action labels** to evaluate the effectiveness of our label generation pipeline
  (See response to **Comment 3** from reviewer **dqkw**, response to **Question 1.2** from reviewer **83of**, **Tables 3,4** and **Section 5.2**)

## Summary of Updates

### Additional Experimental Results
- Updated results on the Bench2Drive benchmark with the force-to-move heuristic reinstated for a fair comparison with SimLingo (**Table 1**).
- Included new ablation studies evaluating the high-level policy and the impact of refined meta-action labels (**Tables 2–4**, **Sections 5.1–5.2**).
- Added open-loop evaluation results on NuScenes (**Table 6**, **Appendix A2**).

### Revised Manuscript
- All updates in the manuscript are highlighted in **blue**.
- Added open-loop evaluation results and expanded implementation details in the **Appendix**.

Please let us know if our responses adequately address your concerns, we are happy to clarify anything further.

---

### Meta-Review · Area_Chair_11Ms · 2026-01-06

**Summary:**

This paper proposes a two stage hierarchical architecture for VLA for autonomous driving. The higher level is done by a VLM, fine-tuned on driving related visual question answering task. This produces both meta actions and reasoning traces about the scene and action. The lower level planner takes the meta action along with scene information and generates the trajectory. There is also a pseudo label generation presented, that can be used for data augmentation in any VLA related tasks.

Reviewers expressed concerns including the following themes:

(1) existing works separate language (meta actions) and control (low level planning) as a system, and that the proposed method does not have a significant lead over existing methods (e.g. SimLingo), or does not fully reproduce (e.g. removal of 'force to move' heuristic), or whether stronger-trained FM backbones are causing improvement versus system design.
(2) data generation is not novel nor described or evaluated strongly.
(3) complexities in frequency and synchronization of the hierarchical system create unaddressed problems.
(4) possible data leakage with speed and turning angle for refinement vs. measurement.
(5) lack of address of Sim2Real gap.
(6) claiming open-loop evaluation, but all tables showing closed-loop eval.
(7) lack of comparison to existing long-tail scenario methods and benchmarks (PARADrive, DiMA, AutoVLA, NAV-HARD)

**Reviewer Concerns:**

The rebuttal made reasonable cases addressing the reviewer concerns as follows:

(1), (6), & (7): further experiments to re-evaluate with the SimLingo force-to-move heuristic, still showing better performance from the proposed method, and comparison to the requested methods on nuPlan open-loop benchmark, showing better performance again from the proposed method.

(2): expanded explanation of data generation method. While not justifying that the data is fully validated, the authors make a case that because the generated data improves performance, whether or not it is perfectly accurate, it is helpful to the method.

(3): addressed in rebuttal as a non-issue, with frequencies provided (5 and 20 Hz), and comments on synchronization capabilities.

(5): sim2real gap is mentioned in rebuttal as a fair concern, with future work proposed, but no direct changes. This is not necessarily within the scope of the contribution, but is a reasonable point towards translating the research to an effective real-world system.

**Reviewer Scores:**

Based on the specific concerns and reasoning of each reviewer, I would expect:

PUZ2: unchanged (2), focus on novelty relative to prior works.
wN4F: increase (5), focus on practical concerns in synchronization, etc.
83of: unchanged (4), focus on data augmentation method novelty and evaluation.
dqkw: unchanged (2), focus on performance relative to prior works.

While the authors provided further experiments and evaluations on requested benchmarks which do support the author claims, it is unclear whether these are sufficient evidence to change the mind of reviewers on main points about the novelty of their method and the marginality of performance gains.

---

### Decision · Program_Chairs · 2026-01-26

Reject